# Diffraction Impact onto Regularized Plasma Channel Formation by Femtosecond Laser Filamentation

Ekaterina Mitina [1,*], Daria Uryupina [1,2], Daniil Shipilo [1,3], Irina Nikolaeva [1,3], Nikolay Panov [1,2,3], Roman Volkov [1,3], Olga Kosareva [1,3] and Andrei Savel'ev [1,3]

[1] Faculty of Physics, Lomonosov Moscow State University, GSP-1, 1-2 Leninskiye Gory, 119991 Moscow, Russia; dasha_uryupina@mail.ru (D.U.); shipilodan-frya@mail.ru (D.S.); irarubik@mail.ru (I.N.); panov_na@mail.ru (N.P.); rv_volkov@phys.msu.ru (R.V.); kosareva@physics.msu.ru (O.K.); abst@physics.msu.ru (A.S.)

[2] V.E. Zuev Institute of Atmospheric Optics SB RAS (IAO SB RAS), 1, Academician Zuev Square, 634055 Tomsk, Russia

[3] P.N. Lebedev Physical Institute of Russian Academy of Sciences, Leninsky Prospect 53, 119991 Moscow, Russia

\* Correspondence: convallaria_lo@mail.ru

**Abstract:** Focused femtosecond beam filamentation after amplitude masks has been studied experimentally and numerically. We deduced conditions (energy per hole, diameter and geometrical composition of holes, focal length) providing for the formation of the regularized bundle of filaments or single on-axis filament at the given pulse duration and beam diameter. We showed that a light channel with small diameter ($\sim$200 μm) and overcritical peak power may be formed well before both the focal distance and the Marburger length, and this channel collapses due to self-focusing and forms the filament. The start position of such a filament can be predicted based on the linear propagation equation, while a more sophisticated non-linear approach that takes into account the Kerr nonlinearity, plasma effects, etc., helps to describe the temporal structure of a filament, its frequency, and its angular spectrum.

**Keywords:** femtosecond filamentation; regularized array of filaments; diffraction; Kerr effect





## 1. Introduction

Filamentation of femtosecond laser radiation in a gaseous medium [1–3] is accompanied by weakly ionized channel formation [4–6]. The transverse size of the plasma channels is about 100 μm and their length reaches several meters [7] or even kilometers [8], while the free electron density is usually in the $10^{15}$–$10^{17}$ cm$^{-3}$ range [9]. This phenomenon has a wide range of applications. In particular, ordered and controlled arrays of femtosecond filaments are of great interest. An electron density wave propagating after a laser pulse and oscillating at the plasma frequency can be a source of terahertz radiation [10]. The use of a controlled array of filaments makes it possible to form an array of THz sources, fulfill conditions for coherent summation of THz radiation from several filaments, and obtain a narrow beam of THz radiation [11–13]. Plasma relaxation and subsequent local heating of the medium results in a cylindrical shock wave formation around the filament [14]. At about several microseconds after the laser pulse passage, it is possible to achieve surrounding gas density perturbations on the scale of several hundred micrometers [15]. In [16], it was shown that, using an array of four femtosecond filaments in air, it is possible to form a dynamic waveguide and transmit a laser pulse (1 mJ, 7 ns, 532 nm) over a distance of $\sim$70 Rayleigh lengths [16,17].

Multiple filaments are observed normally with a stochastic arrangement of individual filaments in the beam cross-section [18–20]. The number of filaments approximately corresponds to the number of critical powers in the laser beam divided by 2–5, depending on the

beam spatial quality (for air and laser pulse duration less then 100 fs $P_{cr} \sim$ 5–10 GW [21]). One needs to control both the spatial distribution of the filaments in the beam cross-section and their longitudinal spatial position for most applications. While the first goal can be achieved by an artificial wavefront modulation of the initial laser pulse, beam focusing is used for the latter goal.

It is possible to create a regular array of filaments placing amplitude or phase masks in the beam [22–30]. Most often, phase masks are used to form ordered arrays of the filaments [16,17,25–27]. But this approach is limited by the self-action effects in the mask material. In addition, phase masks are quite sensitive to intensity inhomogeneities in the beam wavefront, which are always present in high-power laser systems [29,31].

An alternative way is to use amplitude masks or meshes [22–24,28,30,32], which are opaque screens with different configurations of holes. The main drawback here is energy losses introduced by the mask. In [33], it was shown that intensity oscillations on the beam axis arising after beam truncation by the hole due to Fresnel diffraction can be used to control a single filament position inside the glass block moved along this axis. Certainly, here not the intensity changes themselves (since filamentation is governed by the beam power, not intensity) but changes in the beam diameter (which determines the self-focusing distance at the given power) were the cause of the observed behavior. In [24], theoretical analysis was fulfilled for the femtosecond beam filamentation after the rectangular hole. It was shown that energy fluence in the beam cross-section can be predicted using linear diffraction theory, while Kerr nonlinearity comes into play provided the beam is overcritical, i.e., contains too much peak power. Recently, stable arrays of filaments were achieved in a 40 m long atmospheric path by configuring the femtosecond beam properly (namely its diameter and power, number and diameter of holes) [30]. The filament bundle here starts exactly at the distance equal to the distance corresponding to the opening of the first Fresnel zone calculated from the single hole diameter, while the filament spatial distribution follows the initial hole configuration.

In this paper, we have shown experimentally that interplay between the self-focusing, Fresnel difraction and external focusing lengths determines for the given peak power of the femtosecond laser beam not only the filament start position and length but also the filament transverse pattern prescribed by the amplitude mask. In most cases, it is enough to consider the well-known linear diffraction problem of radiation propagation to understand the initial stage of filamentation and to analyze the transverse energy distribution at various distances. This approach makes it possible to predict the transverse structure of multiple filament and plasma channels in both stochastic and regularized modes.

## 2. Experimental Setup

A Ti:Sa laser system with a pulse duration of 55 fs, pulse repetition rate of 10 Hz, wavelength of 805 nm, pulse energy up to 20 mJ, and beam size of 7 mm at FWHM was used to form filaments (see Figure 1). The filament was regularized using amplitude masks—opaque plates with four holes. The effect of the size and location of holes on the formation of a filament bundle was studied. We chose four masks, their parameters listed in Table 1.

**Table 1.** Parameters of amplitude masks. Here, *d* is the distance between the centers of the holes, *D* is the hole diameter, and $F_0$ is the distance corresponding to the opening of the first Fresnel zone.

| Identifier | M0 | M1 | M2 | M3 |
|:---:|:---:|:---:|:---:|:---:|
| Mask type |  |  |  |  |
| *d*, mm | 5 | 3 | 5 | 8 |
| *D*, mm | 4 | 2 | 2 | 6 |
| $F_0$, m | 5 | 1.25 | 1.25 | 11.25 |
| Transmittance, % | 50 | 16 | 10 | 41 |

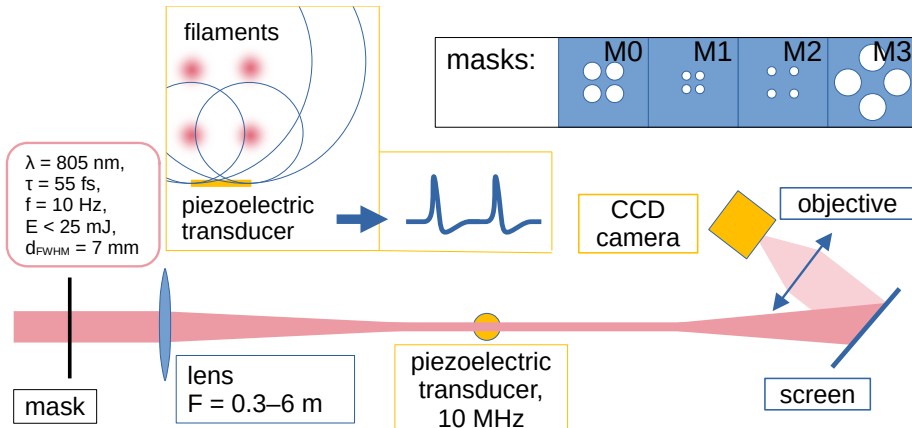

**Figure 1.** Experimental setup. The left inset shows a cross-section of the laser beam with four filaments and the acoustic detector. Acoustic waves of two near and two far filaments give two peaks, which are detected by a detector.

The table shows the diameter of the holes $D$ as well as the distance between the centers of the holes $d$. The fifth row of the table shows the distance from the mask's hole $F_0 = D^2/(4\lambda)$ when the first Fresnel zone exposes at incident radiation wavelength $\lambda = 805$ nm. Each mask cut out a different part of the incident Gaussian beam. The fraction of laser pulse energy transmitted through the holes of the mask is shown in the transmittance row for each mask. While energy after the entire mask determines the behavior of the central filament near the focus of the lens, the energy after one hole determines filament formation after a hole. The beam was focused by a lens with focal length $F = 30$–600 cm placed right after a mask.

Fluence distribution in the beam (radiation mode) after the filament termination was studied. The laser beam was blocked by the white screen, and the image from the screen was transferred to a CCD camera (MindVision MV-UB130GM-T, Shenzhen MindVision technology co. LTD., Shenzhen, China) using an aberration-free lens. If necessary, neutral filters were placed in front of the camera to attenuate the radiation intensity.

Acoustic measurements were carried out in parallel with the radiation mode measurements to determine the parameters of the plasma channel inside the filament. The method [34,35] allows us to resolve the transverse spatial structure of multiple filaments and to estimate the filament diameter and absorbed energy density. A piezoelectric transducer based on a PVDF film with a receiving bandwidth of 10 MHz was placed near the filament at a distance of 2–5 mm. The plane of the acoustic detector film was parallel to the mask's side and the optical axis. The diameter of the working area of the piezoelectric film was 6 mm. The signal from the acoustic detector was digitized by the high-speed two-channel analog-to-digital converter (LA-n1USB, 8 bits, maximum sampling rate 1 GHz, Rudnev-Shilyaev Ltd., Moscow, Russia).

The appearance of a few acoustic peaks in a detected signal reflects the presence of several filaments. A schematic cross-section of the filaments and the acoustic detector in a plane, perpendicular to the optical axis, are presented in the inset in Figure 1. The circles around the filaments show the acoustic waves as they approach the acoustic detector. The acoustic signal of a single filament has a single oscillation of compression and rarefaction. Acoustic waves of two near and two far filaments overlap and form two acoustic peaks. In that case, volumetric energy density increased twice, while the estimated filament diameter remained almost unchanged.

Acoustic maxima were detected and their positions in time were marked in each acoustic measurement. Further, the parameters of the signals that arrived at the same time

in different laser shots were averaged. Thus, it was assumed that a similar arrangement of filaments remained from shot to shot. It was indeed observed in the experiment during the formation of a regularized filament. The diameter of the heat source, the volumetric energy density, and the linear energy density were estimated from the amplitude and width of the peaks [34].

### 3. Results

*3.1. Experimental Results with M0 Mask versus Linear Diffraction Theory*

The effect of focal length $F$ of the lens on the transverse structure of the multiple regular filaments was studied at different laser pulse energies in the first series of experiments. The M0 mask was used (see Table 1). Figure 2 shows radiation modes at the distance of about 1.5$F$ with $F = 30$–400 cm. The radiation mode measurements were made at the distance where the post-filament did not damage the screen. In the first column, the calculated beam mode is presented in the linear propagation regime. The beam propagation was modeled using the [36] package. The mask, lens, and the incident Gaussian beam were set on a 2D array consisting of $512 \times 512$ elements and corresponding to a 10 mm $\times$ 10 mm square in space. Further, using the Rayleigh Sommerfeld method [37], an array was obtained at a given distance from the initial plane.

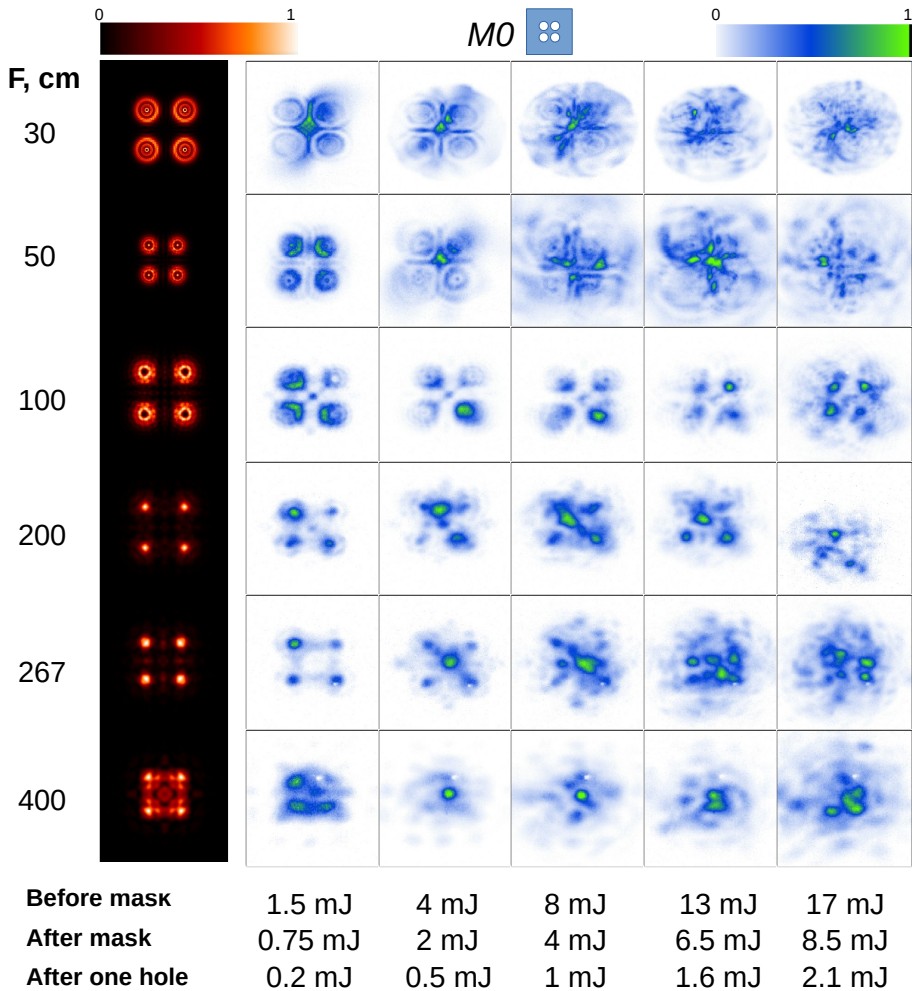

**Figure 2.** Calculated using linear propagation model (1st column) and experimental (other columns), fluence distributions of radiation using lenses with different $F$ at the distance $z = 1.5F$. Different columns for experimental data correspond to the different laser pulse energies, which are indicated in the first line under the figure.

Other columns (2–6) show experimental measurements at different laser pulse energies. Laser pulse energies before the mask, after the mask, and after one hole are indicated below the figure. Our previous studies showed that single filament formation by the unperturbed (without amplitude modulation) laser beam occurs at the initial pulse energy ranges from 2 to 5 mJ [35], which corresponds to the peak power above 40 GW (the critical power of the Gaussian beam in air is 5–10 GW [21]). Hence, the energy range was chosen to trace the beam evolution in three different modes of propagation: (i) the quasi-linear case (energy per hole ∼0.2 mJ, peak power 4 GW), (ii) non-linear case (energy per hole ∼0.5–1 mJ), when the total energy passed through four holes was enough to start a filament, and (iii) high-energy case (energy per hole 1.6–2 mJ), when a single filament was formed after each hole of the mask.

As expected, radiation propagates quasi-linearly at low energy (second column in Figure 2). One can see that calculated and measured modes at the minimum energy are almost identical. An unstructured illumination of the CCD frame was observed at $F = 30$–50 cm and higher energies, which is due to plasma glow from the breakdown region. The bright white intensity maximum was observed along the beam axis at $F = 267$–400 cm and laser energy of 0.5–1 mJ per hole (the third and forth column in Figure 2). The beam corresponding to this maximum originated in the region close to the focus of the lens, did not disappear in the far field, and had low angular divergence. Radiation from the four beams cut out by the mask's holes was also present in the mode, but the brightness of these spots was weaker than that of the central spot. The central beam corresponds to the post-filament [38], which is a continuation of a single filament created from the non-linear merging of the four beams along the initial optical axis (on-axis filament). A small part of the leading edge of a laser pulse, which was weakly affected by the plasma, participates in the post-filament formation. Energy of this part of the pulse should be sufficient to maintain non-linear propagation. In the case of loose focusing, part of the laser beam forms the mode, which can exist over long distances due to balance of the Kerr nonlinearity and diffraction. In the case of tight focusing, the intensity gradient at the focus is high, almost the whole pulse was affected by the plasma, and no post-filament formation was observed. Decay of the central beam into several post-filaments was observed with an energy increase up to 13–17 mJ (the fifth and sixth columns in Figure 2). This is due to an increase in the radiation power in the axial part of the beam up to several critical powers.

We traced the dynamics of plasma channels during the laser beam propagation. The acoustic signal was measured for each $F$ over a wide range of distances $z$ from the lens. Data obtained with the acoustic measurements are plotted in Figure 3 as color scale squares. The color of a dot indicates the volumetric absorbed energy density in a 6 mm zone along the filament. The width along the vertical axis corresponds to the transverse size of the acoustic source and approximately coincides with the diameter of the plasma channel. The grey-scale image shows the calculated cross-section of laser energy flow in the linear mode. Cross-sections were taken in the plane passing through the beam axis and the diagonal of the mask square formed by the holes.

Analysis of the linear beam propagation showed that at the distance $F_f = (1/F + 1/F_0)^{-1}$ (green line in Figure 3, $F_0$—distance of the first Fresnel zone opening without focusing), a narrow intensity maximum is formed containing ∼0.25 of the energy passing through a hole. At the same time, the self-focusing distance of each laser beam after the hole $L_f = (1/F + 1/Z_M)^{-1}$ (blue line), where $Z_M$ is the Marburger length [39] for a Gaussian beam with diameter equal to the hole diameter $D$, is longer than $F_f$. If the maximum formed due to Fresnel diffraction contains enough peak power for self-focusing, it collapses due to self-focusing at a short distance of a few centimeters (due to the small diameter of 100–200 μm; see the table under Figure 3). This collapse produces a plasma channel, and we see it as an acoustic signal (the start of the plasma channel is depicted as an orange line).

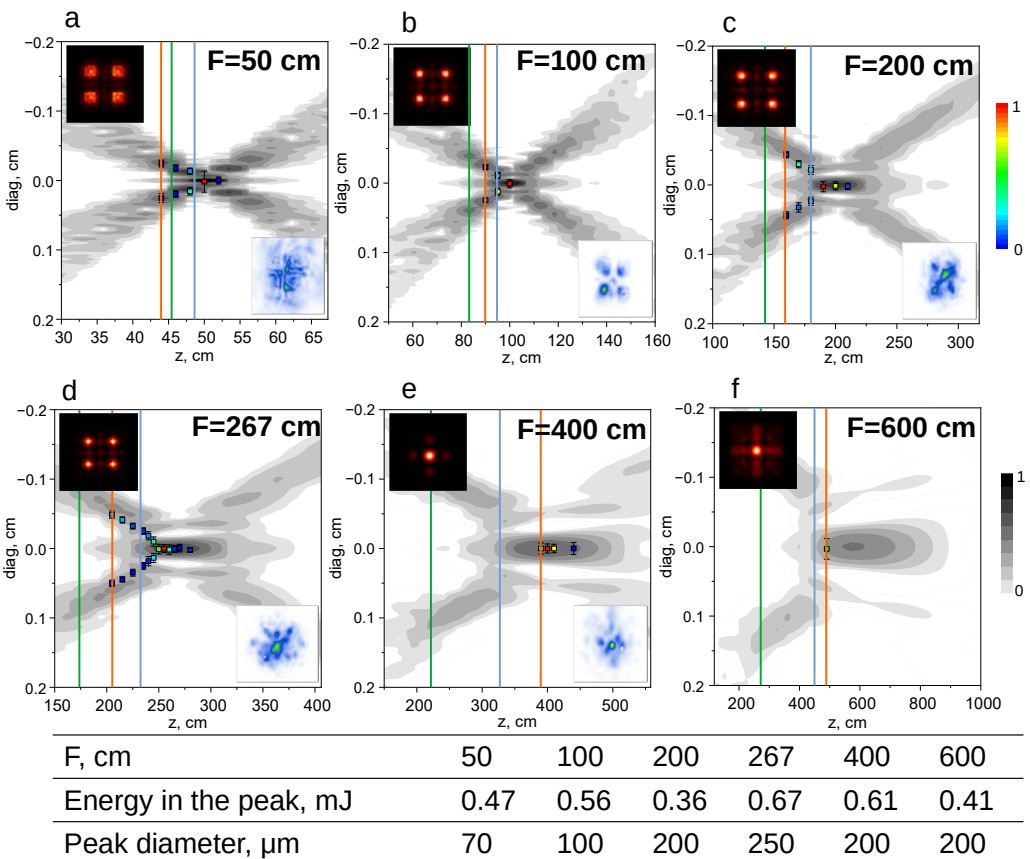

**Figure 3.** Acoustic data (colored squares) for the filament at $F = 50$–$400$ cm, (**a**–**f**) (beam regularization with the M0 mask; laser pulse energy before the mask was 10 mJ). Colors indicate the volumetric absorbed energy density of the acoustic source (plasma channel), while the vertical range shows the source size. The grey-scale images show cross-sections of laser energy flow in the linear propagation regime in the plane determined by the laser beam axis and the mask's diagonal. Laser beam propagates from left to right. Vertical lines mark distance at which acoustic signal is detectable $L_a$ (orange), reduced distance of the first Fresnel zone opening $F_f$ for a single hole in the mask (green), and self focusing distance $L_f$ (blue) (see the text for more details). The insets present beam cross-sections calculated at distances $L_a$ (upper left corners) and measured at the distance $z = 1.5F$ (lower right corner). The image for the $F = 600$ cm was not obtained due to geometrical limitation of the experimental setup. The table below the figure shows calculated energy and diameter of the diffraction maximum arising in the linear propagation mode at distance $L_a$.

Thus, filaments are being formed by each particular beam passed through the hole if $F \leq 267$ cm and energy is high enough (Figure 3a–c). By contrast, the single filament on the optical axis of the initial laser beam is formed for loose focusing ($F \geq 400$ cm, Figure 3d,e). Here, the peak power within the maximum formed by Fresnel diffraction does not exceed the critical one, and the position closer to the focal area diffraction (in particular, ring structure formation) prevents the self-collapse of each beam even at the reduced distance $F_f$. And the single filament forms on the optical axis due to merging of energy flows of the four converging beams.

The laser pulse energy as well as the energy transmitted through each mask's hole affect the location of plasma channels. Figure 4 presents data with laser pulse energies 17 mJ, 10 mJ, and 5 mJ, which correspond to energy per hole of 2.1 mJ, 1.2 mJ, and 0.6 mJ. Peak power sufficient for the filament formation was achieved only after merging four beams in the focal area at the energy of 5 mJ. Here, energy accumulated in a thin 100 μm channel is 0.4 mJ, in accordance with the linear propagation model, and this is enough for the collapse and plasma channel formation. Plasma channels first appear in the side

beams if energy is increased to 10 mJ, and then they converge to the beam center and form a plasma channel at the optical axis. The calculated energy in the central core (200 μm in diameter) of each beam is 0.45–0.6 mJ near the point of the first experimental observation of an acoustic signal. Depending on the energy of the initial beam, the central channel can break up into several closely spaced channels (see Figure 4a).

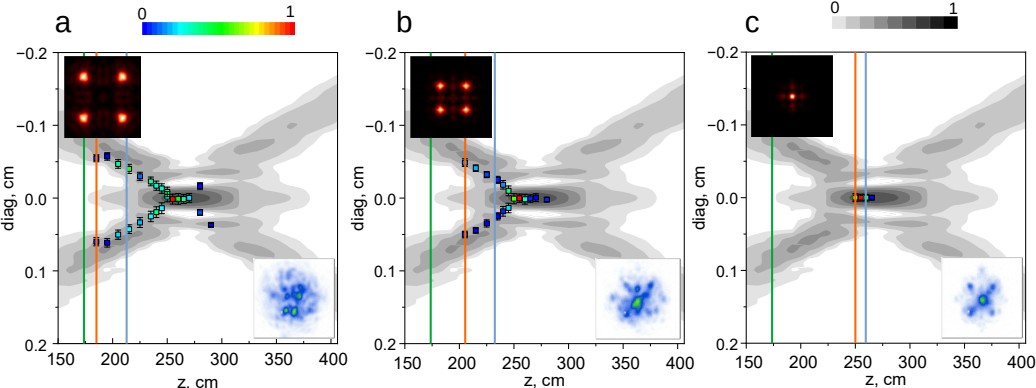

**Figure 4.** Acoustic data and fluence distributions simulated in the linear propagation regime for laser pulse energy 17 mJ (**a**), 10 mJ (**b**), and 5 mJ (**c**) before the mask ($F = 267$ cm, beam regularization with the M0 mask). Other designations are the same as in Figure 3.

The diffraction pattern of the beam after the focus almost repeats the pattern before the focus; however, no filaments are formed in the beams diverging after the focus. This is due to the filament formation. Part of the initial laser energy is spent on ionization and distributes over new spectral components. As a result, the radiation energy in the side beams becomes insufficient for the filament formation.

### 3.2. Non-Linear Simulations of Filamentation with Amplitude Mask

To confirm the dynamics of the plasma channel formation, we carried out numerical simulations. To do this, we used the following non-linear envelope equation (see Equation (73) in [1]):

$$\frac{\partial}{\partial z} A(\tau, x, y, z) = -\frac{i\mathfrak{T}^{-1}}{2k_0} \Delta_\perp A - i\mathfrak{D}A - \frac{2\pi}{c} J, \tag{1}$$

where $A$ is the field envelope, $\tau$ is the time in the frame moving with the pulse group velocity, $x$, $y$ are the transverse coordinates, $z$ is the propagation distance, $k_0$ is the wavenumber at the laser central frequency $\omega_0$, which corresponds to the wavelength of 805 nm, and $c$ is the speed of light in a vacuum. The first term in the right-hand side of Equation (1) describes the diffraction in the paraxial approximation

$$\Delta_\perp = \frac{\partial^2}{\partial x^2} + \frac{\partial^2}{\partial y^2} \tag{2}$$

accounting for the effect of self-steepening

$$\mathfrak{T} = 1 - i\omega_0^{-1} \frac{\partial}{\partial \tau}. \tag{3}$$

The second term represents the dispersion up to the third order, where $\mathfrak{D}$ is the following operator:

$$\mathfrak{D} = -\frac{k_2}{2} \frac{\partial^2}{\partial \tau^2} + i\frac{k_3}{6} \frac{\partial^3}{\partial \tau^3}, \tag{4}$$

and $k_2 = 16.2$ fs$^2$/m, $k_3 = 6.92$ fs$^3$/m are, respectively, the second- and third-order dispersion coefficients calculated for the wavelength of 805 nm [40].

The last term in the right-hand side of Equation (1) describes the nonlinearity in a filament through the material nonlinear current $J$:

$$J(\tau) = \frac{\partial P_{\text{inst}}}{\partial \tau} + \frac{\partial P_{\text{rot}}}{\partial \tau} + J_{\text{abs}} + J_{\text{free}}. \tag{5}$$

Here, the third-order polarizations $P_{\text{inst}}$ and $P_{\text{rot}}$ account for the nonlinear responses of bound electrons [41] and molecular rotations [42], respectively. Since we used the $\sim$50-fs pulse in the experiment, we assumed the Kerr coefficient $n_2$ to be $10^{-19}$ cm$^2$/W [21] for both $P_{\text{inst}}$ and $P_{\text{rot}}$. The last two terms in the right-hand side of Equation (5) describe the effects of the nonlinear ionization. The density of free electrons $N_e$ is governed by the equation

$$\frac{\partial}{\partial \tau} N_e(\tau) = \mathcal{W}\big[A(\tau)\big](N_0 - N_e), \tag{6}$$

where $N_0 = 2.7 \times 10^{19}$ cm$^{-3}$ is the neutral density and $\mathcal{W}[A]$ is the ionization rate calculated according to [43]. The current $J_{\text{abs}}$ determines the ionization-induced absorption of laser energy [44], and the current $J_{\text{free}}$ accounts for the motion of the electrons released in the high-intensity field of a filament [45].

The energy of the laser pulse and the energy transmitted through one hole were 7 mJ and 0.5 mJ, respectively. Figure 5 shows the radiation modes obtained in the calculations (rainbow intensity graphs). At the same distances from the lens, black-red intensity graphs show intensity distributions in the case of linear beam propagation.

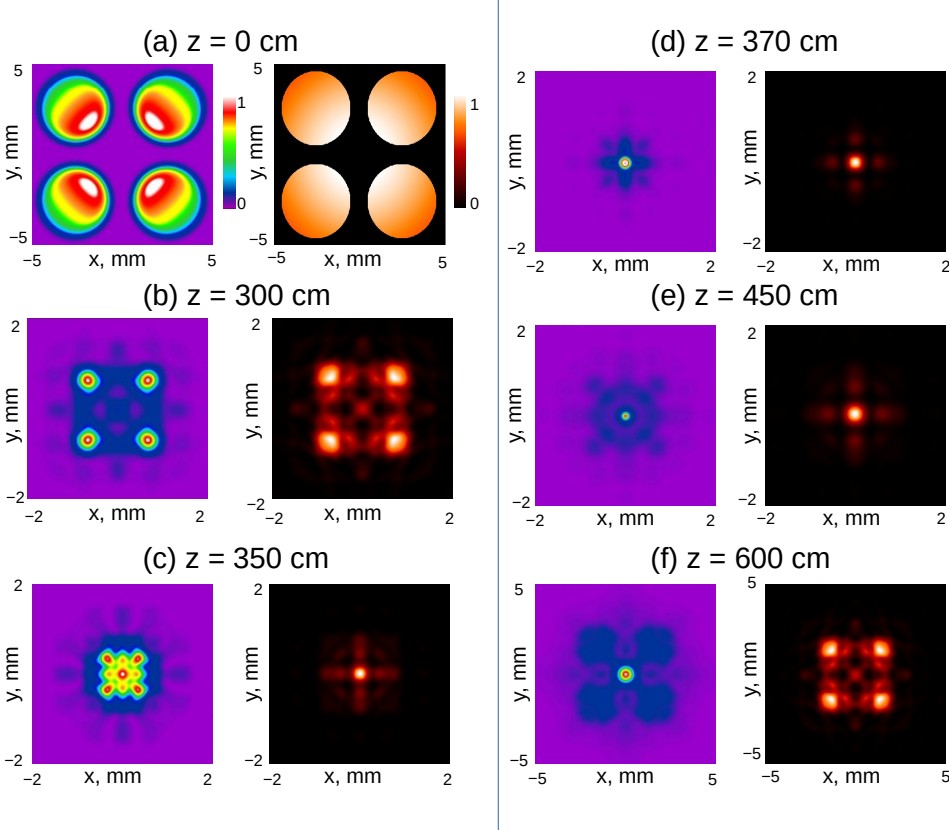

**Figure 5.** Simulated transverse fluence distributions in a beam regularized by the M0 mask (2 mJ energy after the mask, external focusing with $F = 4$ m lens) at different propagation distances $z$. The results of solving the non-linear envelope equation are shown in the first and third columns. The second and forth columns show intensity distributions in the case of linear beam propagation. Distances from the lens are 0 (**a**), 300 (**b**), 350 (**c**), 370 (**d**), 450 (**e**), and 600 (**f**) cm.

The calculated values of electron density in the plasma channel and the radiation intensity along the filament are presented in Figure 6. In reasonable agreement with the experiment (see Figure 3e), free electrons occur in the region of four beams merging near the focus. Peak electron density reaches $\sim 5 \times 10^{16}$ cm$^{-3}$. At the distance of plasma appearance, the diffraction (linear) theory gives estimates for the energy and diameter of $\sim 0.4$ mJ and $\sim 150$ μm (see Figure 5c, linear case), respectively. This agrees well with the experimental data. The plasma channel is maintained over a distance of about 1 m and decays as the radiation intensity in the filament decreases. The length of the plasma channel measured in this case in the experiment was also $(1 \pm 0.1)$ m. The modes obtained as a result of numerical simulation clearly show the formation of a post-filament (Figure 5e) with a small angular divergence. Maintenance of a small divergence occurs due to the balance of the Kerr nonlinearity and diffraction. In this case, the amount of energy in the post-filament beam is about several percent, and the intensity is too low for plasma formation.

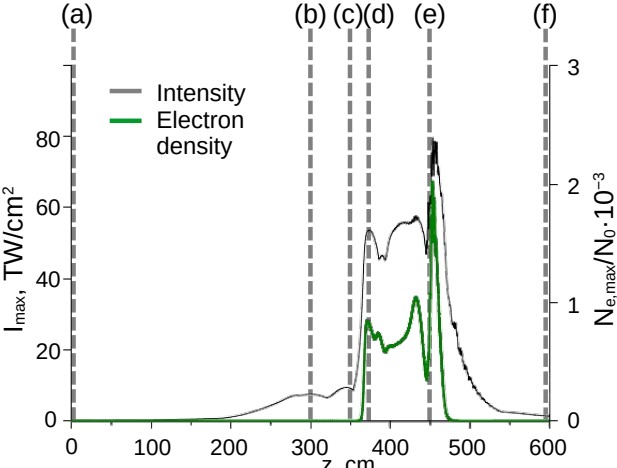

**Figure 6.** Peak intensity of the beam $I_{max}$ (left vertical axis) and free electron density $N_{e,max}$ in the plasma channel (right vertical axis, $N_0 = 2.7 \times 10^{19}$ cm$^{-3}$) as a function of propagation distance $z$ for a regularized beam (M0 mask, laser pulse energy 2 mJ after the mask, focusing $F = 400$ cm). (**a**–**f**) labels correspond to the images in Figure 5, non-linear case.

Thus, the main difference between the data obtained in the linear and non-linear numerical calculations lies in the observation of energy redistribution between different spectral components in time and space, dynamics of on-axis filament, post-filament formation, etc. Energy flow before the focal area follows in general the linear diffraction propagation model, while non-linearity causes sharp collapses of the beam followed by plasma channel production. Nevertheless, the start of the plasma channel can be easily estimated from the linear propagation model.

### 3.3. Effect of Mask Parameters

In the last series of experiments, a structure of plasma channels was studied depending on the mutual position of the mask's holes. Each hole of the mask cuts a circle from the initial laser beam and forms an individual beam. Then, four beams converge at angle $\theta$ depending on the focusing lens. First, we considered the impact of angle $\theta$ on an the axial plasma channel, which formed near the lens focus. Three masks were considered with holes located at the vertices of a square with 3 mm, 5 mm, and 8 mm sides. The hole diameters were chosen to capture as much energy as possible in each quarter of the beam, that is, 2 mm, 4 mm, and 6 mm (see masks M1, M0, and M3 in Table 1).

The lens with $F = 400$ cm was used to form filaments. Figure 7 shows the theoretical linear (first column) and experimental (second to fourth columns) radiation modes after the filament. The laser pulse energy before the mask was 3 mJ (second column), 8 mJ (third column), and 17 mJ (fourth column). Energy after the mask can be obtained using

transmittance, which is presented in Table 1. It can be seen that the modes obtained in the linear mode (second column) accord well with the linear propagation model (first column). A small difference for the M3 mask mode is caused by ellipticity of the initial beam mode and uneven distribution of the radiation energy between the four holes. In the third column, laser pulse energy passing through the four holes is sufficient only for the axial filament formation, but is not enough for filament nucleation in the four converging beams. The highest-energy images (fourth column) correspond to multiple filamentation and, in fact, to the destruction of the filament structure prescribed by the mask. It can be seen in the on-axis filamentation mode (third column) that a single extended filament is formed for the M0 mask. The convergence angle of the beams is $\theta_{M0} = 0.05°$ in this case.

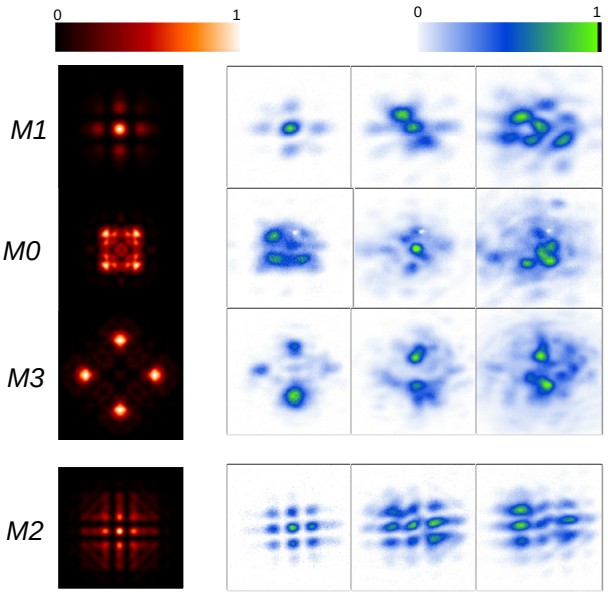

**Figure 7.** Calculated in the linear propagation model (1st column) and experimental (2nd to 4th columns) radiation modes after the filament termination at a distance of 558 cm from the lens with $F = 400$ cm using masks M0–M3. The laser pulse energy before the mask was 3 mJ (2nd column), 8 mJ (3rd column), and 17 mJ (4th column). Energy after the mask can be obtained using transmittance, which is presented in Table 1.

The convergence angle is smaller if using M1 mask ($\theta_{M1} = 0.03°$). This leads to an increase in the split probability of the central filament into several. It happens due to the appearance of nearby beams and interference and non-linear interaction of radiation from individual channels (see diffraction modes in the first column). As a result, the far-field image shows often not one, but two or more filaments. No post-filament from the axial filament was observed in the far zone for the M3 mask with $\theta_{M3} = 0.08°$.

A decrease in the effective numerical aperture and convergence angle leads to an increase in the length of the axial plasma channel (see Figure 8). The post-filament formation was observed only in the first two cases (M0 and M1 masks). It seems that this is due to the much longer light channel formed by diffraction at the optical axes after the focus. Its length can be estimated as ~108 cm for the M3 mask ($\theta_{M3} = 0.08°$), ~170 cm for the M0 mask ($\theta_{M0} = 0.05°$), and ~190 cm for the M1 mask ($\theta_{M1} = 0.03°$).

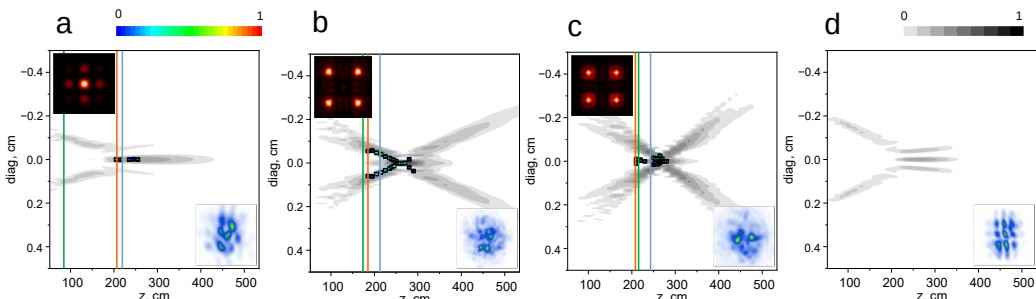

**Figure 8.** Acoustic data and fluence distributions simulated in the linear propagation regime at energy 17 mJ before masks, $F$ =267 cm. Energy per hole was 0.7 mJ with mask M1 (**a**), 2.1 mJ with mask M0 (**b**), 1.8 mJ with mask M3 (**c**), and 0.4 mJ with mask M2 (**d**). Other designations are the same as in Figure 3.

The bottom line in the Figure 7 shows an example of a 3 × 3 array of filament formation using mask M3 with four holes. The spatial arrangement of light channels at the focus differs significantly from the initial configuration of the holes in the mask and is mostly determined by the diffraction and mutual interference of beam radiation. It can be seen that nine extended light channels half a meter long are formed at the focus of the lens (see Figure 7 last row and Figure 8d). The beams are located at a distance of ~100 µm from each other and have approximately the same energy flux. The total radiation energy concentrated in these beams is about 40% of the energy transmitted through the mask. Weak post-filaments formed with laser pulse energy increase in each of these channels (see last row in Figure 7). Small energy passing through holes did not allow us to observe an acoustic signal in this case. However, according to the post-filament radiation mode, this regime was close to the filamentation. As the energy passing through the mask increases up to 0.4 mJ, the radiation from individual channels begins to interfere and interact due to their proximity.

## 4. Discussion

Diffraction of a freely propagating laser beam determines the critical power, and the beam collapses due to self-focusing if its power is overcritical [1–3]. Thus, the laser filament is formed. Furthermore, diffraction governs the low-intensity reservoir spatial dynamics, supporting filament propagation, refocusing, and post-filament formation [38,46]. This well-established picture may change if the initial laser beam undergoes diffraction at an artificial screen with holes, phase mask, etc. While one can find a lot of papers using such a screen to control the transverse distribution of laser power flow (and hence the transverse distribution of filaments), very few works consider the impact of such a diffraction on filament formation itself. It was shown previously by numerical simulations [24] that diffraction at the rectangular hole may create overcritical power fluence at some distance from the hole, and this part of the beam self-focuses, forming a filament.

We have shown that spatial composition, starting point, and length of the plasma channel array formed under filamentation in the air of a focused femtosecond laser pulse with modulated amplitude front depend not only on the number of holes in the amplitude mask and peak power passing through each hole but also on the hole diameter and focal length of the lens. If peak power passed through one hole is below the critical power of self-focusing, but the full peak power passed through all the holes is above critical, we observe single filament formation in the focal area. Its start comes from diffraction-governed nucleation of the power at the optical axes followed by the prompt self-focusing collapse due to the small diameter of the diffraction channel. If the total peak power does not exceed the critical power significantly (2–3 times), a single on-axis filament is formed. Otherwise, it breaks up, and multifilament appears. The single post-filament is formed provided the length of the diffraction channel is long enough, i.e., the convergence angle of the beams after the mask is low (in our study, $\leq 0.05°$), and hence, after the focal area

spatial distribution of the post-filaments followed the Fourier transform of the initial wave front with amplitude modulation and focusing, in this case.

If the peak power passed through one hole is of the order of the critical power, Fresnel diffraction comes into play. It forms the intensity maximum at the axes of the beam passed through the hole, which may contain enough peak power for the prompt collapse (again, the diameter of this diffraction-governed spot is small, 100–200 μm). Typically this spot accumulated 20–30% of the energy passed through the hole, but this also depends on the ratio between the distance of the first Fresnel zone opening and lens focal length $F$. If $F$ is large ($F > 400$ cm in our study), the light spot formed by the Fresnel diffraction is large and contains low energy. Thus, the short filament appears in the focal area, without post-filamentation. Otherwise, each beam produces its own filament and their spatial distribution is prescribed by the mask. Further, these filaments converge in the focal area and form a single or multiple filaments (and post-filaments) at the optical axes, depending on the total peak power. If the peak power is further increased, multiple filaments form both before and/or inside the focal area with a stochastic spatial distribution.

Hence, our experiments clearly showed the crucial role of diffraction for filamentation of a femtosecond beam with amplitude modulation, in contrast to the freely propagating (or loosely focused) Gaussian beams. We suggested the receipt based on the linear propagation and diffraction model as to how to create a regularized filament bundle using amplitude modulation of the femtosecond beam. Numerical simulations of the non-linear envelope equation with terms of dispersion, instantaneous and delayed Kerr nonlinearity, and ionization show that taking into account the complex dynamics of non-linear processes does not improve significantly the assumptions about the position of plasma channels in space, made on the basis of the analysis of diffraction patterns.

**Author Contributions:** Conceptualization, D.U. and A.S.; funding acquisition, O.K. and A.S.; investigation, E.M., D.U., I.N., D.S. and N.P.; resources, R.V.; software, D.S., I.N., N.P. and E.M.; supervision, A.S. and O.K.; writing—original draft, E.M., D.U. and N.P.; writing—review and editing, A.S. and O.K. All authors have read and agreed to the published version of the manuscript.

**Funding:** Russian Science Foundation (21-12-00109). D.E. Shipilo acknowledges the support of the Council of the President of the Russian Federation for State Support of Young Scientists and Leading Scientific Schools (project no. SP-3450.2022.2). I.A. Nikolaeva acknowledges the support of the Foundation for the Advancement of Theoretical Physics and Mathematics BASIS (project no. 21-2-10-55-1).

**Institutional Review Board Statement:** Not applicable.

**Informed Consent Statement:** Not applicable.

**Data Availability Statement:** Data are available on request.

**Conflicts of Interest:** The authors declare no conflict of interest.

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
