# Peer review of "Diffraction Impact onto Regularized Plasma Channel Formation by Femtosecond Laser Filamentation"

_photonics, doi:10.3390/photonics10080928_

Round 1
Reviewer 1 Report
The authors present experimental and theoretical results about the influence of the diffraction on the plasma channel formation during the filamentation of an intense femtosecond laser beam. I find the results very interesting. I recommend the publication of the manuscript but there are a few aspects that the authors could discuss or improve before publishing the manuscript:
* The authors comment several times the possibility of having the Townes mode under some conditions. The Townes mode have some spatial characteristic that helps its identification (its exponential tale). Do the authors find this in the experiment or in the simulations?
* Figure 3 shows two situations in which the filament is formed before the plasma channel is detected (the (e) and (f) cases). Could the authors explain the origin of these result?
* By the way, the last case shown in Fig. 3 (case (f)) does not have the beam cross section. The authors should explain why.
* The authors do not indicate the parameters used in the nonlinear simulations (n2, GVD, TOD, model used for the ionization rates, etcetera). They must indicate all these parameters so the simulations can be repeated by other researchers.
* The authors indicate that “The laser pulse energy transmitted through one hole of the mask is given for each experimental image.” in Fig. 7 but I did not find them.
* Finally, I suggest an improve of English language.
I suggest an improve of English language.
Author Response
Dear Editor,
The authors would like to thank all the reviewers for high appreciation and for the valuable comments, that helped us to improve the manuscript. Changes were marked in red. Below are our answers for the comments.
Reviewer 1
* The authors comment several times the possibility of having the Townes mode under some conditions. The Townes mode have some spatial characteristic that helps its identification (its exponential tale). Do the authors find this in the experiment or in the simulations?
>>Usually the beam indeed forms the Townes mode in the filament, and it was observed in our numerical simulations as well. We did not investigate experimentally an exact transverse shape of the beam in the filament, so we removed ”Townes mode” from the text.
* Figure 3 shows two situations in which the filament is formed before the plasma channel is detected (the (e) and (f) cases). Could the authors explain the origin of these result?
>>The gray-scale image in Fig.3a-f shows the linear simulation of the beam propagation not accounting for self-focusing, etc. Using the acoustic method (colored squares) we revealed the real spatial position of the filaments. Formation of a light channel without plasma before the filament was not observed.
* By the way, the last case shown in Fig. 3 (case (f)) does not have the beam cross section. The authors should explain why.
>>We do not present an image of the beam cross section for the 3f case because the experiment was carried out in the laboratory room and it was not possible to take measurements at a distance of 1.5 F (8-9 m) from the laser pulse compressor output window without additional beam reflections. We mentioned this in the revised paper.
* The authors do not indicate the parameters used in the nonlinear simulations (n2, GVD, TOD, model used for the ionization rates, etcetera). They must indicate all these parameters so the simulations can be repeated by other researchers.
>>We thank the Reviewer for this recommendation. We added the detailed description of the nonlinear model.
* The authors indicate that “The laser pulse energy transmitted through one hole of the mask is given for each experimental image.” in Fig. 7 but I did not find them.
>>We apologize, this phrase remained from the previous version of the manuscript. Information about the pulse energy before the masks has been added to the picture caption and to the text.
* Finally, I suggest an improve of English language.
>>We have edited the text once again and tried to improve English language.
Reviewer 2 Report
This manuscript studied the filamentation of femtosecond beam with amplitude modulation by a mask with holes. The influence of diffraction effect was discussed through comparison with linear propagation mode. This manuscript has certain reference value for the method of generating filament arrays using spatial modulation and is worth publishing. I just have two comments:
1. The physical mechanism of the diffraction effect on the formation of filaments should be appropriately analyzed. In my opinion, filaments are generated by self focusing, but diffraction still dominates the distribution of low intensity energy backgrounds, and filaments will be generated in local strong regions of the energy background?
2. The material of this manuscript is a little bit complex and hardly to read, and I suggest to simplify it appropriately.
Author Response
Dear Editor,
The authors would like to thank all the reviewers for high appreciation and for the valuable comments, that helped us to improve the manuscript. Changes were marked in red. Below are our answers for the comments.
Reviewer 2
1. The physical mechanism of the diffraction effect on the formation of filaments should be appropriately analyzed. In my opinion, filaments are generated by self focusing, but diffraction still dominates the distribution of low intensity energy backgrounds, and filaments will be generated in local strong regions of the energy background?
>>Classical approach about the formation and propagation of laser radiation in the filamentation mode really suggest that the filament is formed as a result of beam collapse due to self-focusing of the high-intensity part of the radiation on the beam axis. At the same time propagation of energy reservoir around the filament is determined by diffraction. In this paper, we consider the propagation of laser radiation after passing through a mask with several holes. In this case, the formation of a more complex diffraction pattern with local maxima and minima inevitably occurs. This leads to a situation where the start of a filament can be determined not by self-focusing, but by a local increase in intensity due to diffraction on a hole. To clarify this point more, we have added a new paragraph to the discussions section.
2. The material of thе manuscript is a little bit complex and hardly to read, and I suggest to simplify it appropriately.
>>With a simultaneous change in the mask configuration, focusing conditions and laser pulse energy, we observed many processes that occur in the beam. Single filament and post-filament formation on the beam axis, formation of an array of filaments and plasma channels were observed. Therefore the manuscript may become incomplete if not to mention some of these processes. At the same time, we have tried to make the presentation of the article a little clearer. In particular, figures 3, 4, 8 are rotated by 90 degrees so that the direction of radiation propagation becomes more familiar, explanations for some physical quantities used in numerical simulations and a new paragraph in the discussion section were added. Some changes have also been made throughout the text.
Reviewer 3 Report
In the paper by E.Mitina, D. Uryupina, et al. focused filamentation of the laser beam with a mask is concidered. The conditions of the filamentation is of great both academic and practical interest. The paper successfully combines experiments and simulations. I would like to mention high quality of the paper and clear presentation of the obtained results.
Author Response
Dear Editor,
The authors would like to thank all the reviewers for high appreciation and for the valuable comments, that helped us to improve the manuscript. Changes were marked in red. Below are our answers for the comments.
In the paper by E.Mitina, D. Uryupina, et al. focused filamentation of the laser beam with a mask is concidered. The conditions of the filamentation is of great both academic and practical interest. The paper successfully combines experiments and simulations. I would like to mention high quality of the paper and clear presentation of the obtained results.
>>The authors would like thank the Reviewer for high appreciation.
Reviewer 4 Report
In the manuscript entitled “Diffraction impact onto regularized plasma channel formation by femtosecond laser filamentation,” the authors have studied the effect of multi plasma formation in air and its interactions by the simultaneous analyses with two distinct methods: one optically and other acoustically.
The authors have studied the effect of four beam filamentation including the plasma formation threshold, the beam geometry and size. In order to produce these four beam, the authors have chosen to use a simple mask with four holes with different geometry. Lens with different focal length were also used to control the laser irradiance. The experimental results obtained were compared with one obtained with numerical simulations. In my opinion, the results obtained in the manuscript is very interesting for a broad audience and can be accepted as it is but, it can be improved if the authors provide extra clarifications.
For example, it seems to me that the four focused beam configuration obtained using a mask with four symmetric holes provide a nonlinear interaction of these four beams in the air at zero delay time. Am I right? If so, there are several third-order nonlinear effect among these four beams such as self-diffraction and four-wave mixing. Can you comment about it?
Is it any white light continuum generation during the filamentation?
In my opinion, the figure of the acoustic data (Fig. 3, 4 and 8) can be more natural and close to the experimental configuration if are plotted rotated by 90 deg, i.e. horizontal axis is z.
Author Response
Dear Editor,
The authors would like to thank all the reviewers for high appreciation and for the valuable comments, that helped us to improve the manuscript. Changes were marked in red. Below are our answers for the comments.
Reviewer 4
For example, it seems to me that the four focused beam configuration obtained using a mask with four symmetric holes provide a nonlinear interaction of these four beams in the air at zero delay time. Am I right? If so, there are several third-order nonlinear effect among these four beams such as self-diffraction and four-wave mixing. Can you comment about it?
>>Both in the experiment and in simulations based on the nonlinear envelope equation, we did not observe any nonlinear effects associated with the cross influence of individual beams on each other. This paper presents experimental data only on plasma channels location and the transverse mode of radiation in a filament, and this does not provide complete information on the spectral composition of radiation. These study is underway in our group, but it seems not good to overload the current paper by additional part.
Is it any white light continuum generation during the filamentation?
>> Yes, we observed supercontinuum generation in all the beam regimes where acoustic signals were observed.
In my opinion, the figure of the acoustic data (Fig. 3, 4 and 8) can be more natural and close to the experimental configuration if are plotted rotated by 90 deg, i.e. horizontal axis is z.
>>Thank you for your comment. We rotated the images
Reviewer 5 Report
The paper is devoted to the study of the femtosecond beam filamentation process. The paper's novelty lies in using amplitude masks of various configurations to create different diffraction patterns. Due to this approach, stable filamentation structures can be created, which may be interesting for a number of applications.
The results obtained in the article are not in doubt and have a high scientific significance, and all the data are presented clearly.
But I would recommend the authors add a description of the procedure for optical-acoustic measurements of the spatial structure of the filament, despite the fact that only one link is given. The paper can be published in its current form.
Author Response
Dear Editor,
The authors would like to thank all the reviewers for high appreciation and for the valuable comments, that helped us to improve the manuscript. Changes were marked in red. Below are our answers for the comments.
Reviewer 5
But I would recommend the authors add a description of the procedure for optical-acoustic measurements of the spatial structure of the filament, despite the fact that only one link is given. The paper can be published in its current form.
>>We presented description of the acoustic method and its applications in a set of papers before. In particular the paper [Uryupina, D. S., et al. (2016) Laser physics letters 13(9), 095401.] is devoted to the development of the wideband acoustic method for determining the parameters of the thermal source of a filament acoustic signal. We have added an extra paper where this method is applied and where a description of the method application is also given.